# The Advances in Computer Vision That Are Enabling More Autonomous Actions in Surgery: A Systematic Review of the Literature

**DOI:** 10.3390/s22134918

**Published:** 2022-06-29

**Authors:** Andrew A. Gumbs, Vincent Grasso, Nicolas Bourdel, Roland Croner, Gaya Spolverato, Isabella Frigerio, Alfredo Illanes, Mohammad Abu Hilal, Adrian Park, Eyad Elyan

**Affiliations:** 1Departement de Chirurgie Digestive, Centre Hospitalier Intercommunal de, Poissy/Saint-Germain-en-Laye, 78300 Poissy, France; 2Department of Surgery, University of Magdeburg, 39106 Magdeburg, Germany; roland.croner@gmail.com; 3Family Christian Health Center, 31 West 155th St., Harvey, IL 60426, USA; vincentgrasso@gmail.com; 4Gynecological Surgery Department, CHU Clermont Ferrand, 1, Place Lucie-Aubrac Clermont-Ferrand, 63100 Clermont-Ferrand, France; nicolas@surgar-surgery.com; 5EnCoV, Institut Pascal, UMR6602 CNRS, UCA, Clermont-Ferrand University Hospital, 63000 Clermont-Ferrand, France; 6SurgAR-Surgical Augmented Reality, 63000 Clermont-Ferrand, France; 7Department of Surgical, Oncological and Gastroenterological Sciences, University of Padova, 35122 Padova, Italy; gaya.spolverato@gmail.com; 8Department of Hepato-Pancreato-Biliary Surgery, Pederzoli Hospital, 37019 Peschiera del Garda, Italy; isifrigerio@yahoo.com; 9INKA-Innovation Laboratory for Image Guided Therapy, Otto-von-Guericke University Magdeburg, 39120 Magdeburg, Germany; alfredo@surag-medical.com; 10Unità Chirurgia Epatobiliopancreatica, Robotica e Mininvasiva, Fondazione Poliambulanza Istituto Ospedaliero, Via Bissolati, 57, 25124 Brescia, Italy; abuhilal9@gmail.com; 11Anne Arundel Medical Center, Johns Hopkins University, Annapolis, MD 21401, USA; apark@aahs.org; 12School of Computing, Robert Gordon University, Aberdeen AB10 7JG, UK; e.elyan@rgu.ac.uk

**Keywords:** artificial intelligence surgery, autonomous actions, computer vision, deep learning, machine learning

## Abstract

This is a review focused on advances and current limitations of computer vision (CV) and how CV can help us obtain to more autonomous actions in surgery. It is a follow-up article to one that we previously published in *Sensors* entitled, “Artificial Intelligence Surgery: How Do We Get to Autonomous Actions in Surgery?” As opposed to that article that also discussed issues of machine learning, deep learning and natural language processing, this review will delve deeper into the field of CV. Additionally, non-visual forms of data that can aid computerized robots in the performance of more autonomous actions, such as instrument priors and audio haptics, will also be highlighted. Furthermore, the current existential crisis for surgeons, endoscopists and interventional radiologists regarding more autonomy during procedures will be discussed. In summary, this paper will discuss how to harness the power of CV to keep doctors who do interventions in the loop.

## 1. Introduction

Unlike AI “in” surgery which consists of radiomics, enhanced imaging analysis and decision making in the pre- and post-operative period, artificial intelligence surgery (AIS) is becoming known as the potential fusion of current surgical robots with artificial intelligence (AI) that could theoretically lead to autonomous actions during surgery [1]. AIS is different from AI in other medical fields because surgery by definition is an interventional field. As time goes by, other interventional fields are increasingly becoming fused with some clinicians able to do surgery, endoscopy and interventional radiologic procedures. Because of this perpetual evolution, we chose surgery as the umbrella term to incorporate all interventional fields. Surgeons, endoscopists and interventional radiologists (IR) have historically relied on their sense of touch and vision, but since the minimally invasive revolution sight has come to the forefront. This is probably best seen by the observation that all robotic surgeons who use complete surgical systems that require a console (da Vinci Surgical System, Intuitive Surgical, Sunnyvale, CA, USA), only have visual information to go by and must operate without any haptics/sense of touch [2].

This review article will address the current advances made in the field of computer vision (CV) as it pertains to autonomous actions during surgery. Additionally, it will discuss the existential crisis for surgeons as to whether or not the safest and most effective way towards more advanced autonomy in surgery should require less or more physical distance between surgeons and their patients. Fundamentally, should we be striving for true robotic autonomy during interventional procedures, are collaborative robots (cobots) the safest way forward or will it be a combination of both? Although robots and computers may be able to effectively “see” through traditional concepts of CV such as instrument priors and motion analysis to create the third dimension, non-visual data points such as the interpretation of audio signals and light intensity readings may also be used. This review is a follow-up to a previously published article that, in addition to CV, also expanded upon the other pillars of AI such as machine learning (ML), deep learning (DP) and natural language processing as they pertain to autonomous actions in surgery [3].

## 2. Methods

Search terms included combinations of autonomous actions in surgery/autonomous surgery/autonomous surgical robots with CV, ML, DL, augmented reality (AR), segmentation, phase recognition or navigation. Additional search terms included less traditional concepts of CV such as instrument priors and audio-haptics. Articles discussing autonomous actions in interventional healthcare disciplines were then sought. The term “surgery” was utilized in the manuscript to encompass the fields of surgery, endoscopy and interventional radiology. The fields of surgery studied included open and minimally invasive approaches. Minimally invasive approaches were defined as encompassing endoscopic and robotic-assisted approaches. Endoscopic approaches were defined as including, but not limited to laparoscopic, thorascopic-assisted and arthroscopic procedures. Only studies carried out or meant for human subjects were considered. Studies that were not in English were excluded. Virtual reality (VR) was not included in this review as it was not considered precise enough in its current form to offer safe autonomous actions during interventions.

Articles that were included were chosen and approved by the first 2 authors (A.G. and V.G.). This review article is meant for medical doctors that perform interventional procedures. The review attempts to familiarize doctors with the basic concepts of artificial intelligence with an emphasis on the most relevant advances in CV that are enabling more autonomous actions during interventional procedures. Review articles and studies about simulating interventions were excluded.

## 3. Results

This review article was registered on PROSPERO on 6 June 2022 and is pending approval. The PRISMA checklist was used to format and organize this article. A total of 1687 articles were identified by searching PUBMED, GOOGLE SCHOLAR and REASEARCHGATE (Figure 1).

A total of 1499 articles were off topic and were excluded. An additional 67 were duplicates. Of 121 articles initially identified, an additional 33 were not in the English language. Of the 88 articles remaining, 3 could not be retrieved. Of these, an additional 13 were found not to describe an autonomous action, leaving 72 articles for this review.

### 3.1. Console Surgical Robots

The manufacturer, Intuitive Surgical, has the most experience in the world. The first human instance where the da Vinci was used in 1998, and the first robot was available to buy since 2000. The da Vinci has three components: a console, the laparoscopy tower and the robotic arms. Unlike the Versius robot, all of the robotic arms are on one base weighing approximately 700 kg. Alternatively, each robotic arm of the Versius is on a separate base weighing only 100 kg. Because of this weight difference, the da Vinci is much stronger, which is fundamental when dealing with inflammatory tissue. However, this added strength may lead to more iatrogenic injuries when the da Vinci is used, especially early in one’s learning curve. Because of the fact that the da Vinci robot is on a solitary base that weighs more, it has become the dominant force in pelvic surgery. Unlike surgery in the abdomen that can easily be carried out from the side of the patient or by operating in-between the legs, during pelvic surgery there is no comfortable position for a surgeon to stand. As a result, the da Vinci will probably always be the dominant type of robot for pelvic surgery. Namely, remote surgery carried out at a console, which is also known as tele-manipulation. Unlike its competition, the da Vinci also has an energy device and powered stapler that can fit onto the robot.

The main advantage of other complete console surgical systems (Versius, CMR, Cambridge, UK) is that it costs almost ½ the price of the da Vinci to buy. In addition, the disposable instruments that are needed for every procedure also cost approximately ½ the price. The fact that each robotic arm has a separate base also increases the positions that the arms can be placed. Theoretically, this can lead to more complex robotic approaches and procedures. However, early in a surgeon’s career, this may lead to difficulties with set-up and could lead to some surgeons becoming frustrated with the robot and abandoning its use altogether. Another potential advantage to the Versius that was mentioned above is that the fact that the Versius robotic arms weigh less and can thus create less force may lead to fewer iatrogenic injuries, at least, early in a surgeon’s learning curve. Technically, the Versius robot has the ability to enable haptics; however, the computer is so sensitive that it senses the surgeon’s resting tremor rendering its activation more deleterious than beneficial. 

Future robots that are being developed to involve the use of handheld robotics and enable the surgeon to maintain contact with their patients. These robots may cost even less than the Versius system, but they are currently not available on the market [1]. The two different visions for robotic surgery, namely, tele-manipulation with a surgeon sitting at a console or handheld robotics with surgeons standing right at the operating room, are crucial to understanding which is the best method towards more autonomous actions in surgery. To better understand the significance of these two approaches, the difference between automatic and autonomy must be understood. 

### 3.2. Automatic vs. Autonomous Actions

In contrast to autonomous actions, automatic actions involve no interpretation of data prior to the action. Autonomous or intelligent actions involve the interpretation of sensor data and/or utilization of ML algorithms to decide whether or not to do the proposed action (Figure 2). Additionally, autonomous actions may enable devices to alter how actions are carried out. In essence, ML enables devices to make decisions for which it was not necessarily programmed. These two different types of actions can maybe be best understood by comparing two kinds of commercially available straight gastrointestinal anastomotic (GIA) powered staplers. One of these powered GIA staplers (Echelon Stapler, Johnson & Johnson, New Brunswick, NJ, USA) does not have any sensor or interpretation capabilities, it is just activated and fires. Alternatively, another powered straight GIA stapler has a sensor and once activated it will ascertain the thickness of the tissue before firing (Signia™ Stapling System, Medtronic, Dublin, Ireland). If the tissue in the stapler’s blades is too thick it will not attempt to fire, or once the tissue becomes too thick it will stop. Additionally, if the tissue is not thick enough, it will not even begin stapling. It is hard to imagine that this type of sensing technology is a form of AI that results in decreased staple line and anastomotic leaks.

There is a profound lack of understanding in the surgical community as to what constitutes autonomy and what AI even is. AI is not a zero sum game and to help explain the varying shades of grey of AI and autonomy, the concept of strong and weak AI has been created [3]. Strong AI being the traditional understanding of AI where a robot is independent and essentially does everything. Using this definition, it is easy to see why so many surgeons are skeptical as to the viability of full autonomous actions in surgery. 

As researchers have grappled with the intricacies of autonomy, six levels of surgical autonomy have been designated. Notably, these six levels follow closely the schema developed for self-driving cars [4]. Level 0 is no autonomy with levels 1 through 4 being examples of weak AI/autonomy. More specifically, level 1 corresponds to the tele-manipulation that we see with current complete robotic surgical systems. Level 2 involves some interpretation of data by the machine and corresponds to the intelligent-powered stapler discussed above. Level 3 autonomy is when the surgeon has several autonomous actions that can be chosen. Energy devices that have multiple sealing settings that independently determine the time of sealing via algorithms and indicate the failure or success of sealing could be an example of this example of weak AI. 

Level 4 autonomy would be when a device actually makes medical decisions but is still controlled by a physician. Percutaneous ablation of arrhythmias with devices placed inside the heart by interventional cardiologists, but robots on cardiac catheters that sense electrical myocardial misfirings and then autonomously do ablations are a modern day example of this level of autonomy [5]. A true example of strong AI or level 5 autonomy where a robot or device acts completely independently does not exist in surgery; however, an example does exist in healthcare. An automatic implantable cardioverter-defibrillator (AICD) autonomously diagnoses an arrhythmia and potentially gives a life-saving shock treatment to the patient without any input from a human [6].

### 3.3. Computer Vision (CV)

Enabling a robot to have the ability to see, CV, has become the main obstacle to the development of truly autonomous actions in surgery. Organ segmentation, the ability to differentiate organs and structures within organs, and phase segmentation, the ability for computers to recognize which part of the procedure is being carried out, are remarkably complex tasks to teach surgical robots (Figure 3). In the non-healthcare fields, this limitation of modern-day CV has been resolved by having humans control some aspects of a robot’s actions. These collaborative robots or “Cobots” are increasingly used in the business world to enhance productivity during manufacturing. The current limiting factors for a robot to function autonomously are elucidated when see the trials and tribulations of getting a robot to reliably and securely help a neurologically affected patient to drink a glass of water without the aid of another human [7]. 

### 3.4. Machine Learning (ML), Deep Learning (DL) and Computer Vision (CV)

As discussed above, ML incorporates both weak AI (autonomy levels 2–4) where the mind’s ability to make decisions is only partly mimicked, and strong AI (level 5) where the more complex decision functions of the human brain are emulated. Deep learning (DL) algorithms were created to more closely imitate the anatomical structure of the brain in an attempt to attain higher decision-making capacity. DL falls within the umbrella of ML and employs multiple neural networks (NNs) structures such as dee, recurrent and convoluted NNs in an attempt to attain a higher level of sophistication during decision making, resulting in the potential for more complex actions. It is through the more recent improvements in DL mathematical and geometrical models that have resulted in reliable and functioning CV.

### 3.5. Detecting and Classification of Objects in Images

Ultimately, it is possible that the greatest contribution of laparoscopic surgery to healthcare will be that it resulted in the ease of a computer being placed in-between the patient and surgeon [8]. This innovation has enabled images to be classified on either static or video images. This image classification has been carried out via the labeling of images and the creation of training sets that can be tested and adjusted as needed until the computer can reliably and effectively act upon CV data.

Many advancements in object detection have come from attempts at facial recognition, such as techniques created by Michael Jones and all Viola (Figure 2). Their approach utilizes a ML algorithm known as a Haar Cascade that has a classifier that learns by interpreting positive and negative images. Notably, existing literature suggests that Haar cascades can outperform convolutional neural networks (CNNs) in some specific tasks such as localizing surgical instruments [9]. Model libraries containing images of surgical instruments have to be used to train the algorithms to differentiate the different instruments. Vast databases containing videos of surgical procedures such as for colon surgery [10], gallbladder surgery [11] and obesity surgery [12] already exist and are used to analyze different methods of object detection [10].

Advanced deep-learning-based methods such region-based CNN (R-CNN) [13], single shot detectors (SSD), region-based fully convolutional networks (R-FCN), YOLO (You Only Look Once) [14] and others in recent years showed superior performance over other traditional methods in detecting and recognizing objects from images and video footage. Each of these methods can be utilized successfully in certain medical situations and are subject to the context. For example, YOLO is considered one of the fastest object recognition methods; however, faster R-CNNs may have better performance [15]. Typical successful application of these methods include the work in for the detection and localization of tools in robot-assisted surgery (RAS) [16]. Detailed review of the application of DL-based methods for autonomous surgery can be found in the journal *Artificial Intelligence Surgery*.

Automated bounded boxes, which are boxes that are placed around objects in images are utilized to permit the detection of objects. The more objects that there are in an image, more boxes will be created and appear (Figure 4). The necessary libraries are downloaded and coding programs can be devised that can learn how to classify and identify the objects of interest. For surgical instrument detection, the algorithm is first taught how to identify one instrument and then how to detect and identify various surgical instruments. This is carried out by giving each different instrument a specific and precise label. The labels are then converted into a digital output that can be read by a neural network that can ultimately differentiate each surgical instrument. In general, 80% of the image library is required to train the model with only 20% used for testing the model. If the accuracy is not adequate, then the mistakes need to be studied and the algorithm altered to enhance the NN.

### 3.6. Current Optics Used in Minimally Invasive Surgery

Three different colors, red, green and blue (RGB), that are visible to the human eye are used to make color images and each pixel has a specific value and a well-defined range. Digital endoscopes and laparoscopes mimic each of these three RGB light wavelengths. Additionally, strong white light is needed to create illumination. The ability of the human eye to perceive things is currently being enhanced during surgery via biophotonics such as indocyanine imaging and narrow band imaging (NBI) [17]. A form of functional imaging, indocyanine imaging works on the principle that some molecules that can be absorbed in the human body can re-radiate fluorescent light. During hepatobiliary surgery, specialized hardware can detect fluorescent indocyanine that binds with bile, which is a yellow-colored pigment that peaks in the blue-green spectrum. When complex algorithms are used to interpret this information, specialized images can be generated that can enhance the visual identification of anatomical structures, tumor location and lymph node drainage and blood flow. This blue-green spectrum is also ideal for the generation of NBI, which can also augment the identification of blood vessels resulting in the enhanced detection of cancers and pre-cancers [18].

Surgical spectral imaging (SSI) is another modality that is still in the experimental phase and includes multi-spectral imaging (MSI) and hyper-spectral imaging (HSI). As opposed to more traditional imaging that utilizes the RGB spectrum, spectral imaging also uses a third data point for each individual pixel. However, unlike a traditional geometrical third dimension, spectral imaging relies on a data point derived from the reflectivity of the tissue being imaged and currently only relies on images that have two spatial dimensions. The resulting data cube is more specific than standard two-dimensional imaging and this enhanced imaging is believed to be superior at the differentiation of organs and has been utilized in space exploration and marine studies [19]. MSI measures spectral bands on a factor of 10 and HSI measures spectral bands on a scale of 100. HSI “fingerprints” have been created in an animal model that studied the differentiation of up to 20 different internal tissues and organs with an accuracy superior to 95%. It is hoped that HSI could lead to computers ultimately being able to accurately and reliably differentiate organs in real-time [19]. The difficulty of organ identification pales into comparison of the algorithms needed to give computers the ability to differentiate the different parts of an organ, this task is known as organ segmentation. Once you factor in the need for a third spatial dimension for the realistic development of autonomous surgery, the computational challenges that have to be overcome become clearer. Because of this complexity, we will focus on two-dimensional static images first, then discuss the interpretation of video and lastly touch upon some of the issues of grappling with three-dimensional spaces.

### 3.7. Semantic Segmentation

Semantic segmentation, also known as image segmentation, is stratified into four overall phases [3,15,20,21]. Phase one is classification, phase two is localization and classification, phase three is object detection and phase four is considered true semantic segmentation. Classification is when something is correctly identified within an image. Localization and classification are when an object is localized correctly within a bounding box. Object detection is the proper identification and differentiation of something in a picture when there are multiple objects within the image. Semantic segmentation occurs when the computer can properly localize and identify objects even when the object is over-lapping other things within the picture. This is accomplished by having every pixel in the image classified and differentiated based on predictions, this process is termed deep prediction. Semantic segmentation is needed for the computer to determine how instruments are oriented and will be needed so that the trajectory of instruments can be predicted.

Currently, limitations in adequate segmentation are perhaps the main rate-limiting step for future innovations and progress. Although virtual bronchoscope navigation (VBN) for radial end-brachial ultrasound and ultra-thin bronchoscopy (UTB) has been available since 2008, clear advantages have been limited. For example, a randomized-controlled multi-center trial did not demonstrate a statistically significant superiority of VBN compared to unassisted UTB. On the other hand, when nodules in the distal 1/3 of the lung or nodules that were not seen on fluoroscopy were analyzed separately, the patients that had VBN had a statistically significant improvement in nodule detection [20]. Other centers also showed a non-significant tendency for superior diagnostic yield in patients that underwent UTB with VBN when compared to standard UTB alone (47% vs. 40%, respectively). However, of note, when cases with optimal segmentation were analyzed separately, diagnostic yield was found to be significantly better, with diagnostic yield increasing to a rate of 85% [21].

In perhaps some of the most interesting examples of available AI in healthcare, algorithms utilizing clinically functioning semantic segmentation has enabled real-time diagnosis of suspicious lesions during endoscopy and even assistance with classification of anatomy during laparoscopy [22,23,24,25,26]. Various CNNs have been created to successful allow for this level of semantic segmentation and include the deep learning algorithms ENet, UNet, SegNet and ErfNET. When a data set of laryngoscopy images were analyzed, improved segmentation of laryngeal tissue was noted when UNet and ErfNet were used, but better efficiency was noticed when ENet was used [22]. The reality that alternate algorithms are better at solving different aspects of CV elucidates the degree of complexity that is necessary for computers and ultimately surgical robots will need to see as well as the human eye.

### 3.8. Instance/Video/Surgical Segmentation

Instance segmentation involves the differentiation/identification of an object, when more than one of the same objects is in an image frame, for example, when two identical laparoscopic graspers are in the same image. In short, a more sophisticated level of segmentation is needed. The complexity of this task becomes clearer when we understand that the different parts of the same instrument must also be accurately identified, this type of segmentation is termed multi-class segmentation. Linknet and TeranusNet are neural networks that have been used to accomplish this task [27,28]. Color pixel analysis combined with analysis of reflective capacity or texture to create an additional degree of differentiation has been used with these DL algorithms to create functioning instrument segmentation during robotic-assisted and laparoscopic surgery [29]. 

Trying to differentiate the various organs and then parts of each organ is another particularly difficult task that is encompassed in multi-class segmentation. The complexity and difficulty of this task is easier to understand visually. In this example, a patient undergoing a laparoscopic radical cholecystectomy with excision of her common bile duct for an invasive gallbladder cancer (Figure 5a) also happened to have a vascular anomaly, specifically, a replaced right hepatic artery (Figure 5b). One bounded box is placed over all of the arteries and the various arterial branches are not correctly labeled and it is not clear that the patient has a right hepatic artery coming off of the superior mesenteric artery instead of the common hepatic artery (Figure 5c). To account for this anomaly, an additional layer of segmentation, organ segmentation, is required. Notably, this extra layer of identification/segmentation should probably be referred to as vascular segmentation (Figure 5d).

Another method to allow for instrument segmentation consists of a pre-trained encoder and UNet neural network decoder that utilizes nearest-neighbor interpolation [30]. As opposed to standard laparoscopy, robotic-assisted surgery that uses complete robotic surgical systems has the capacity to record and interpret data regarding instrument location into DL algorithms. These types of data are referred to as instrument priors can enhance a robots ability to accurately identify robotic surgical instruments and even their respective parts [31]. As instrument segmentation has evolved, researchers have shown that algorithms can be developed that can even correctly interpret datasets that are publicly available and not only datasets obtained locally under controlled environments. Fortunately, even images only annotated every 10 frames per second have been shown to be adequate to obtain accurate instrument segmentation [32]. Instrument segmentation has even been realized in real-time via the utilization of multi-scale feature fusion [33].

The development of surgical segmentation has been pioneered by the group from Strasbourg, France, with early studies focusing on cataract surgery and minimally invasive gallbladder removal [34]. Although the primordial limitation is the difficulty in obtaining sufficient amounts of datasets of the procedures, an even more time-consuming limitation is the fact that surgeons are needed to annotate these vast repositories of video datasets. In an effort to overcome this limitation, the team from France has proposed the utilization of temporally constrained neural networks (TCNN), which are semi-supervised methods that may facilitate the process of annotation and thus surgical segmentation. This is possible by analyzing both temporal and spatial signals via auto encoder networks [34,35]. The goal of this review article is to highlight advances in CV, future review articles will focus on sensors and sensor fusion [36].

### 3.9. Navigation, Augmented Reality (AR) and Mixed Reality (MR)

It is apparent that future surgeons will be astonished by how we had to operate while being effectively blind. This is easy for older surgeons to understand when we remember how we used to access central lines blindly or the days of endoscopy prior to endoscopic ultrasound or solid organ surgery prior to intra-operative ultrasound. Some lesions in solid organs are only seen on cross-sectional imaging and only in limited instances have surgeons been able to operate with the aid of more sophisticated imaging. Augmented reality or AR has been available in the research setting for over a decade, with commercially available devices only recently coming onto the market. AR that can locate and differentiate blood vessels from other tubular structures such as the bile ducts, lymphatic vessels and ureters is a difficult task that can be carried out with AR, but this is only reliable and accurate with non-mobile and non-elastic structures such as bones and retroperitoneal structures. AR uses cross-sectional imaging that is usually obtained pre-operatively, but theoretically, a similar level of organ segmentation could be obtained with the abovementioned segmentation techniques (Figure 6A,B).

Because of the limitations of the algorithm’s ability to take into account flexibility (Figure 7), AR has been adopted more diffusely in neuro- and orthopedic surgery. This is an active area of research with the top teams currently engaged in the registration of deformable organs, estimations of the deformation and tracking in real-time of the degree of deformation of organs [37,38,39,40]. Workflows also exist for tumors of the gynecological tract (Figure 7) [38,41], urinary system (Figure 8) and hepatobiliary system (Figure 9) [42,43,44,45,46]. Some studies have combined AR with VR to create a mixed reality (MR); however, current technology does not permit the safe creation of autonomous actions during surgery using this approach and is limited to guiding the surgeon [47].

Other limitations of autonomously functioning robots in healthcare have been on movements created by the actions of the heart and lungs. This obstacle was overcome in the 1990s by the CyberKnife system that used sensors placed directly on the patient’s chest so that damage to neighboring healthy tissue during external beam radiation could be minimized [1]. Although autonomously driving cars and prostheses for artificial vision have many of the same problems as CV during procedures, the fact that the human body is elastic, deformable and frail with certain parts that are in constant motion render CV during surgical, endoscopic and interventional radiological procedures infinitely more complex [48].

Prior to the development of more autonomous actions during interventional procedures, in addition to a reliable and rapid ability for the computer to identify organs and instruments, the robot also needs to know how to safely navigate inside the body. This magnitude of this task is best comprehensible when we study the complexity of navigating only a static image. Virtual reality (VR) has been touted as a modality that could give surgeons useful information on anatomic anomalies, location of vascular structures and other critical structures resulting in improved and safer navigation. One key benefit of VR is that for things such as the simulation of operations and virtual endoscopy, lower image resolution is used, as a result, less data are analyzed and computations of algorithms can be carried out faster [49]. Realistically however, it is doubtful that less CV resolution will be a viable option during something as complex as autonomous surgical actions. Nevertheless, VR may have a role during the autonomous screening of lesions during diagnostic/screening endoscopies.

A form of unsupervised DL known as deep Q reinforcement learning (RL) is value-function based and is being used to enable autonomous endoscope navigation. Appropriately, researchers have first attempted to render autonomous the first task of endoscopy, the initial intubation. During diagnostic laparoscopy, the initial intubation is usually carried out in a relatively blind fashion by using feel. By using images obtained during this initial part of the bronchoscopy and not any images obtained pre-procedure, researchers were able to use deep Q RL in a CNN known as DQNN (deep Q reinforcement learning neural networks), to safely get the bronchoscope into the breathing tube [50]. Q-learning RL calculates the optimal policy or skill, which is the best value for a pre-defined task. This form of RL is based on the concept that an exact value for an action can be determined that corresponds to a precise situation and environment. A carefully determined balance must be walked between exploration and exploitation for this to function. The importance of the task being carried out successfully is paramount during exploitation; however, during exploration a higher degree of error is tolerated by the algorithm so the robot/computer can learn better, resulting in superior future actions [51].

## 4. Challenges and Open Areas of Research in CV and Artificial Intelligence Surgery

It can be said that deep-learning-based methods have significantly advanced research and development in the area of robotic surgery, across various tasks including objects detection and recognition, classification, navigation and construction of 3D representation of the environment. However, one of the key requirements for successfully implementation of DL-based methods is the availability of large volumes of carefully annotated datasets. In the surgical environment, this can be very expensive and labor-intensive tasks [3,52]. More importantly, in some scenarios, it can be even impossible to acquire such data, for example, when you consider the need for estimating depth information for endoscopic surgery images, which is an important task to facilitate navigation in a surgery setting using a robot or a semi-autonomous device. In the deep learning era, if we can obtain large volumes of good quality videos with the corresponding depth maps, then such a task may be very possible [53]. 

However, this is quite impossible in a surgical setting due to the dynamic and diverse nature of such an environment. A closely related challenge is the ability to register 3D images (once constructed) especially in such a surgical, dynamic and changing environment (organs deformation, light variations and others). Additionally, there is a need to develop methods that can generalize better to unseen scenarios and this might require training models to learning from multimodal data sources, as well as creating new methods for fusing and integrating these data modalities to inform the learning process. It has to be also said that the quality of the data continues to challenge state-of-the-art deep learning models. Existing literature suggests that deep learning models’ performance, despite their superior performance over human’s across many tasks, becomes similar to human performance on lower-quality data [54]. 

### 4.1. Dexemes/Surgemes/Situation Awareness

Surgical maneuvers have been classified into smaller gestures termed surgemes, using hidden Markhov models (HMMs), data from these various movements during tele-manipulation robotic surgery can be registered and analyzed [55]. These HMMs have been found to be able to stratify surgeons according to where they sit on the learning curve by analyzing the precise tissue–tool interactions [56]. Additionally, analysis of torque and force data has been able to successfully classify skills when using the complete robotic surgical systems [57]. Subsequent studies calculated kinematic measurements with more dimensions via linear discriminant analysis, and Bayes’ classifiers were used to increase to four dimensions so that the quality of surgical gesture segmentation could be maximized [58]. 

Accuracy of surgical maneuver segmentation was further enhanced by again using HMMs to analyze even smaller parts of surgical maneuvers called dexemes [59,60]. What this approach to the analysis of surgical gestures shows us is that the computer can, in essence, see and process surgical movements that the human body and eye may not even notice or register. The ability of computers to analyze the movements of surgical robots that are tele-manipulated and that robotic devices create autonomously may be as relevant as the analysis of more studied and traditional forms of CV. 

For instance, an anastomosis of the small intestine would be divided into several steps or surgemes, specifically, the placement of the two ends of intestine next to each other, the placement of sutures to fix the orientation of the two ends together, the creation of the two enterotomies, the creation of the stapled anastomosis with linear staplers or with sutures and if indicated the mesenteric defect closure. Dexemes would be the different hand gestures of the tele-manipulator that are needed to perform each one of these various steps/surgemes. A sum of the different dexemes would make one surgeme and the sum of a set of surgemes would make up an entire procedure such as cholecystectomy, minor liver resection all the way up to more complicated procedures such as a total gastrectomy or major hepatectomy [61].

Other examples of AI surgemes/dexemes are robots that can autonomously create cochleostomies and others that can carry out knot tying [62,63]. Another interesting effort to create an autonomous action was the development of an independently functioning robot that was created to be able to autonomously accomplish a peg and ring task via the analysis of motion trajectories that can adapt in real-time [64]. To accomplish this task, the research team used the da Vinci Research Kit and dynamic movement primitives and answer set algorithms to develop a working framework. This framework works by imitating the movement of each dexeme, but with the added enhancement of situation awareness that enabled the robot to constantly adapt to a changing environment with new obstacles to overcome. The authors believe that this may be the first time that situation awareness was combined with an autonomous robotic action with a documented ability to correct errors and recover in real-time in a surgical environment [64]. 

### 4.2. Phase Recognition

Phase recognition can help in hospitals by managing workflows, for example, other healthcare providers can be alerted to ongoing progress in the endoscopy suite or operating room. It can help alert providers during interventions to abnormalities such as hemorrhage or missed injuries. Theoretically, phase recognition may be easier to obtain than some of the other tasks that are mentioned above. This is because it may be easier to create algorithms that can teach the computer to identify steps during cholecystectomy such as clipping of the cystic duct and differentiate them from visually different steps such as dissecting the gallbladder off of the liver bed. Alternatively, differentiating more similar steps such as cystic artery clipping from cystic duct clipping may prove more challenging especially when anatomic variations are present. Furthermore, more complex cases that have more anatomic variability, locations in the abdomen and steps will stress the system even more. For example, during total colectomy, all four quadrants of the abdomen are involved, and during pancreatic head resection, the surgeon may go back and forth between operative fields and by definition, between operative phases rendering useful autonomous phase recognition a difficult task indeed.

A review article on ML published in the *Annals of Surgery* in 2021 noted a significant increase in publications on AI with 35 articles dedicated to resolving the task of phase recognition [65]. Research in this field is not limited to HMMs, there is also a lot of interest on the viability of artificial neural networks in creating reliable and effective phase recognition. Most of the datasets analyzed are from feature learning of videos of surgical procedures with the annotation of instrument utilization carried out manually by a trained expert, often a surgeon in training or fully trained surgeon [65]. 

As mentioned above, large datasets are used for the detection of objects, but they are also fundamental to the development of phase recognition models. Some datasets of minimally invasive cholecystectomy are publicly available and include, EndoVis workflow challenge dataset, Cholec8 and MICCAI 2016 that are available for training and testing [65]. In summary, standardized procedures such as cholecystectomy and sleeve gastrectomy are ideal minimally invasive surgeries to study when developing phase recognition AI architectures, but for less standardized operations such as hepatic-pancreatic and biliary or colorectal surgery this will take much more sophisticated algorithms. Currently, the main obstacle is the relative dearth of usable surgical videos of these procedures and the fact that these procedures are significantly longer when compared to more routine procedures such as gallbladder removal and restrictive bariatric surgery.

### 4.3. Robotic-Assisted Surgery and Autonomous Actions

The daunting task of realizing more autonomous actions in robotic-assisted surgery is highlighted by the fact that the first complete robotic surgical system on the market (da Vinci Surgical System, Intuitive Surgical, Sunnyvale, CA, USA) was initially conceived as a tele-manipulator that was supposed to allow surgeons to remotely operate on soldiers injured during conflicts via open surgical techniques. At the same time, as the research team began to realize the complexities involved in making robotic tele-manipulation viable, safe and effective, the laparoscopic revolution was in full swing and the company completely shifted focus to become a tool for minimally invasive surgery [66]. To date, AI in so-called complete robotic surgical systems is limited to simulator training and evaluating surgical skills, and the only true differences to standard laparoscopy are the added degrees of freedom and ability to intermittently control a third arm [51]. Both of these historical advantages are now also available during so-called traditional laparoscopy [3].

Just as with the human body, CV is not only dependent on visual cues, but also information on position and proprioception. Similarly, one of the most useful forms of data for analyzing surgical movements, skills and tasks is motion analysis. Before the era of DL and HMM, surgical skill could only be evaluated by real-time observations of surgeries or by reviewing recorded videos of surgical procedures [67]. Due to the vast amount of data created during procedures that can take several hours, a useful way to hasten the analysis and evaluation of surgical videos is the utilization of time action analysis. This technique only analyzes data from non-continuous fixed intervals so that analysis can be carried out quicker. It is likely that this type of solution may not be safe for the creation of more complex autonomous actions in surgery such as dissection, but may be useful for simpler tasks such as the placement of surgical clips. Because of motion priors analysis with HMMs, algorithms have been developed that can automatically calculate gesture and skill metrics without using any visual information [51]. Next steps will be to combine data from both non-visual time action analysis with visual data, but the massive amount of data that has to be analyzed is currently a significant barrier.

To date, one of the most exciting examples of autonomous actions in surgery is the autonomous creation of a gastrointestinal anastomosis [68]. Accomplished via the Smart Tissue Autonomous Robot (STAR) in an animal model, this robot was able to perform an autonomous, but supervised surgeme via an open approach. To do this, the computer was able to create useable CV by combining a plenoptic three-dimensional tracking system with near-infrared fluorescence (NIRF) imaging. The fact that this was successfully carried out is especially impressive because this task was implemented on soft tissue, which is flexible and malleable. The research team also found that the robot had better skill metrics when compared to surgeons with a minimum of 7 years of experience.

The STAR has a vision system, surgeon interface, robotic arm and a force sensor. To generate a working CV, the cut intestine has to first be injected with biocompatible NIRF markers until a “point cloud” is fashioned around the edges of the cut porcine intestine so that the robot can know where to place the sutures. It is clear that even though this robot can carry out some actions autonomously, it is still completely reliant on a human placing fluorescent markers for working CV to be a reality. Nevertheless, this shows that once a robot can see sufficiently, even complex surgical tasks such as sutured gastrointestinal anastomoses can be carried out autonomously. It should be noted that although the robot’s performance was deemed to be superior to humans, this was based on movement criteria and not on clinical criteria such as stenosis or anastomotic leak rates. This emphasizes the dangers of evaluating autonomous actions in surgery on short-term mathematical movement criteria alone.

The aspiration of blood is a crucial task during surgery and the CV needed to accomplish this task autonomously is surprisingly difficult [69]. The initial obstacle that researchers had to overcome was the reliable and accurate detection of the blood contour. After this was carried out, a mask R-CNN method was used to create a robotic prototype that was able to aspirate blood. Known as the Blood Removal Robot (BRR), this system needs a robotic arm, two cameras, an aspirator, a suction tip and tubing. The BRR has been used in an animal model created to simulate skin and then craniotomy. The best trajectory for the aspiration is calculated using a CNN; however, the robot does not yet have the ability to “see” any other instruments in the surgical field and cannot yet take into account the existence of more than one area of bleeding [69].

Dissection around scar tissue created by inflammatory tissue from benign or malignant disease, infection or previous disease is the most difficult task for surgeons and will certainly be the most daunting task for AIS. Companies have already begun to gather as many data of surgical gestures and movements created by the arms of complete robotic surgical systems [70]. It is important to know that every time a procedure is carried out with the da Vinci robotic system, the motion data of the robotic arms are being recorded and transmitted to the manufacturer. Engineers and computer scientists hope that the sheer quantity of these data will permit the generation of functioning algorithms that will ultimately result in more autonomous actions by surgical robots [71]. Although it is tempting to wonder whether or not this is possible, perhaps this is not the most important question. However, maybe we should be asking ourselves which is the best way forward, more specifically, are complete robotic surgical systems with tele-manipulation really the best way forward? Maybe the surgeon needs to be kept in the loop by keeping the surgeon at the bedside and developing more handheld collaborative robots [6,72].

#### Haptics vs. Audio-Haptics

AI has become a reality thanks to advances in ML and DL, which is a kind of ML approach using multi-layer neural networks. Both of these models can be used in supervised and unsupervised tasks. CV, which is the field of AI that enables computers to interpret images, in turn has become a reality because of advances in DL. However, similar to the human body, the concept of sight and interpretation of digitized visual inputs by computers does not fully define all the ways that a computer can interpret its surrounding, gather information and act effectively and safely on that information. In addition to analyzing pixel data, computers can also incorporate non-visual data such as motion analysis and instrument priors. These types of additional data highlight the potential significance of haptics in the future development of more autonomous actions in surgery. An insightful description of the utilization of random forests to track microsurgical instruments during retinal surgery was published in 2018 in a book entitled *Computer Vision for Assistive Healthcare* [73].

The analysis of sound or audio-haptics may also be an interesting tool in the quest for more autonomy. Computer scientists and engineers have been studying sound waves to see if algorithms can be developed that will give surgical robots even more information [74,75,76]. Because sound waves may require less memory, it is hoped that the analysis of sound waves will give computers more sensitive ways to obtain the relevant information with less data crunching. This may enable more useful information for the robot to have and less time lost during the analysis, resulting in more AI that can actually be used in real-time. Additionally, these types of data could give pixel data another dimension and theoretically improve computers and robots ability to safely perform autonomous tasks [76,77]. Alternative techniques devised to allow for the differentiation of tissues during surgery involve the utilization of electrical bio-impedance sensing and analysis of force feedback, but are still in the prototype phase [78].

## 5. Discussion

Regardless of the varying forms of haptics that may augment a computer’s ability to effectively see and work safely, digitized visual information is still the dominant data type that is being studied in the field of CV. Initial steps in CV involve object detection and classification that is being carried out with either AR with analysis of cross-sectional or sonographic imaging or with bounded boxes and real-time segmentation. Different types of segmentation include, but are not limited to, instance, semantic, organ, multi-class, video and surgical. Neural networks are the main models used to accomplish both AR and segmentation. The existence of standard laparoscopic and robotic-assisted platforms that can enable surgeons to better visualize anatomy and highlight anomalies is to date the most credible aspect of CV. The hope is that as imaging of the body improves, not only will robots be able to navigate the human body more safely, but that surgeons will also benefit from the AR. Improved imaging will allow both the robot and surgeon to operate more safely, which should translate into better patient outcomes and encourage further development of increasingly less invasive procedures.

The path towards autonomous actions in surgery will take many innumerable small steps before we arrive, but this should not generate disillusionment with AI, in fact, an improved understanding of the principles of AI should result in more surgeons embracing its future promise. By separating surgical procedures into its smaller surgemes and even dexemes, any advancement will be more easily appreciated as the only way towards improved outcomes for patients. Even though prototypes of phase recognition and situation awareness already exist, we must constantly ask ourselves, what is truly the best way to AIS? Is surgery with a complete robotic surgical system necessitating tele-manipulation and the surgeon in no contact with the patient, or as in the manufacturing world, are handheld cobots the best way forward? What we do know is that the technology and know-how exist and are growing at an exponential rate. The ability for computers to see is constantly improving. Unfortunately, unless more surgeons learn about AI, we must begin to wonder if surgeons will also continue to improve at what they do best, or if they will become a victim of their own hubris.

## Figures and Tables

**Figure 1 sensors-22-04918-f001:**
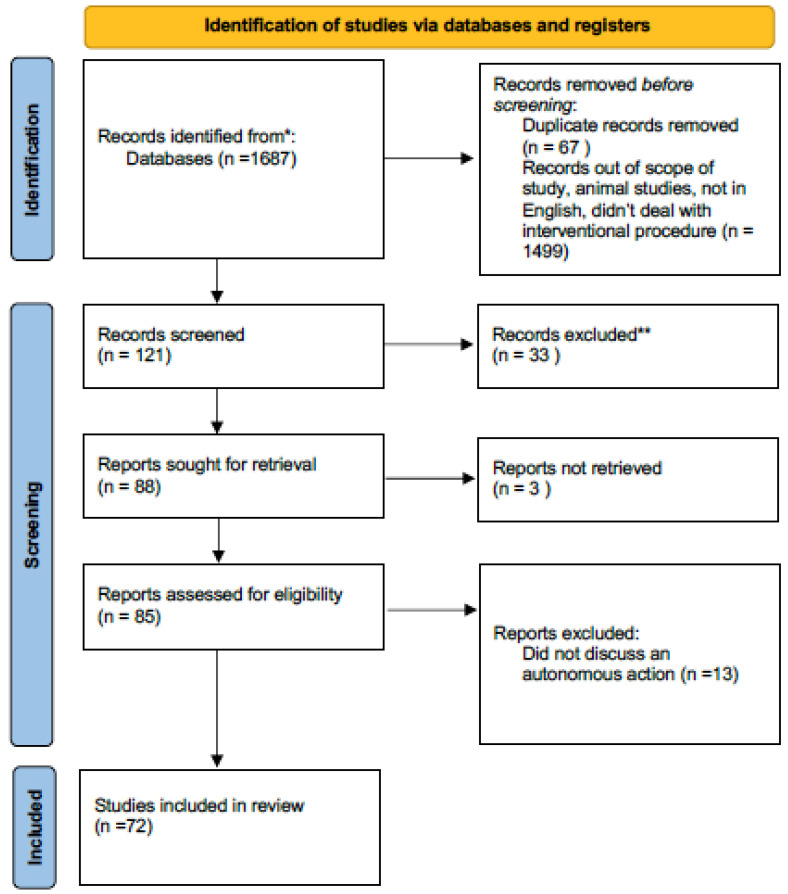
PRISMA flow diagram for this review article. * Pubmed ** did not discuss autonomous actions in surgery.

**Figure 2 sensors-22-04918-f002:**
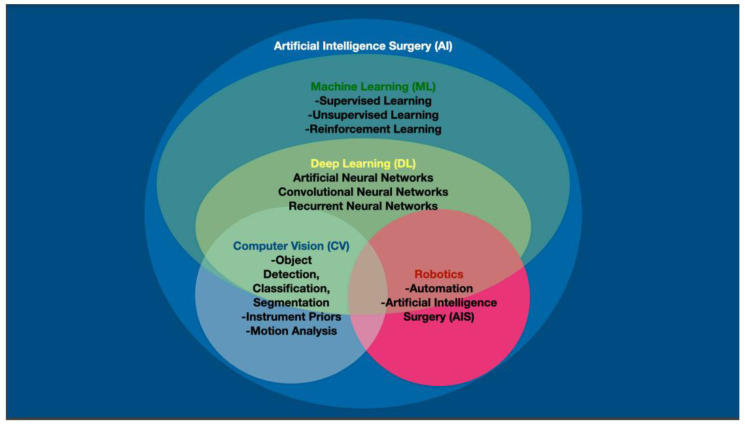
Artificial intelligence (AI) in computerized visualization involves machine learning (ML), which encompasses deep learning (DL). Computer vision (CV) is made possible through the neural networks of DL. Computers and robots may be able to attain autonomous surgical actions through a combination of traditional CV, but also through instrument priors, motion analysis and other non-visual data points.

**Figure 3 sensors-22-04918-f003:**
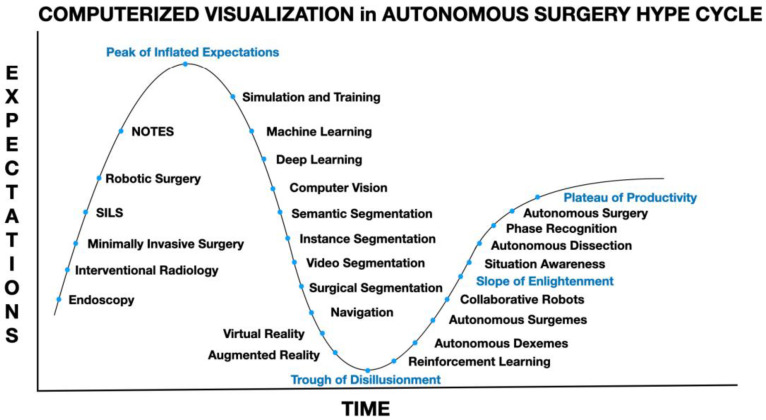
Computerized visualization in artificial intelligence or autonomous surgery hype cycle with potential steps necessary in-between the peak of inflated expectations, trough of disillusionment, slope of enlightenment and plateau of productivity.

**Figure 4 sensors-22-04918-f004:**
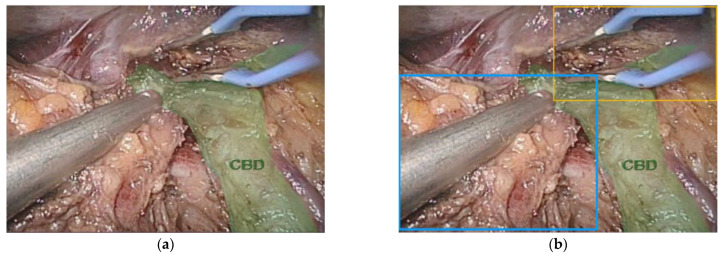
(**a**) Image of the common bile duct (CBD) (green) during minimally invasive major liver resection and (**b**) bounded boxes around surgical instruments. Notice that the left and right hepatic ducts are not labeled.

**Figure 5 sensors-22-04918-f005:**
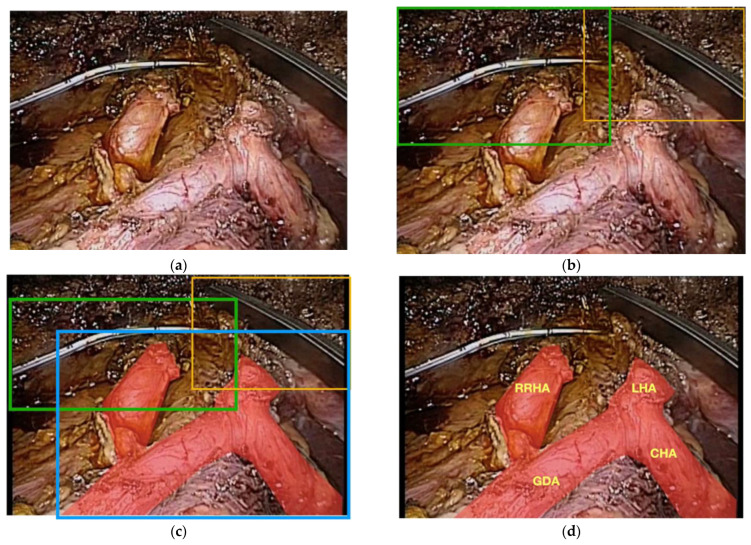
(**a**) Raw footage of the portal triad during a minimally invasive radical cholecystectomy and common bile duct excision for a patient with gallbladder cancer; (**b**) bounded boxes showing instance segmentation of the surgical instruments; (**c**) bounded boxes of instruments and entire arterial supply without multi-class segmentation of the different arteries; (**d**) patient has a replaced right hepatic artery that is not identified by lower-level segmentation.

**Figure 6 sensors-22-04918-f006:**
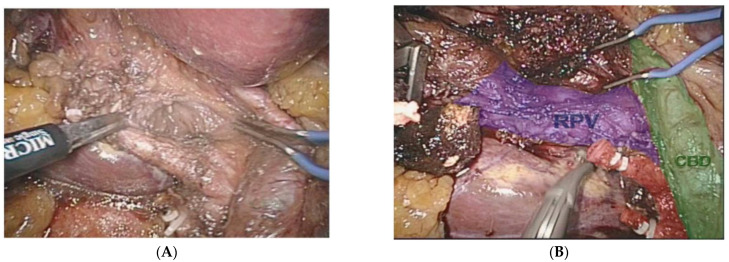
(**A**) Augmented reality (AR) during a minimally invasive extended right liver resection. (**B**) Organ segmentation is possible with the aid of pre-operatively obtained images. Common bile duct (CBD) in green, right portal vein branch (RPV) in blue, clipped and cut anterior and posterior right hepatic arteries in red.

**Figure 7 sensors-22-04918-f007:**
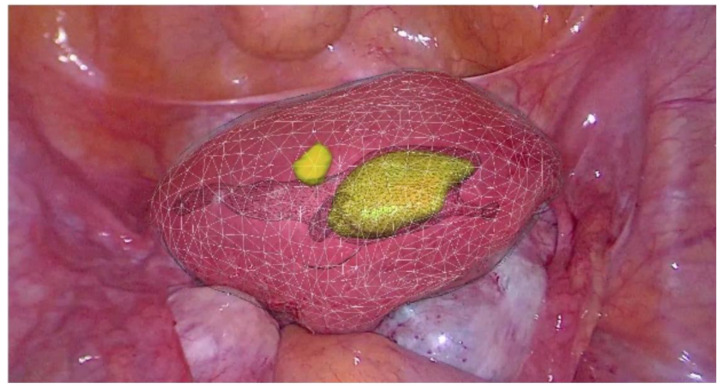
Real-time augmented reality (AR) for gynecology. Uterus is mobile and deformable (white) and intra-uterine tumors (yellow) (EnCoV, SurgAR, Clermont-Ferrand, France).

**Figure 8 sensors-22-04918-f008:**
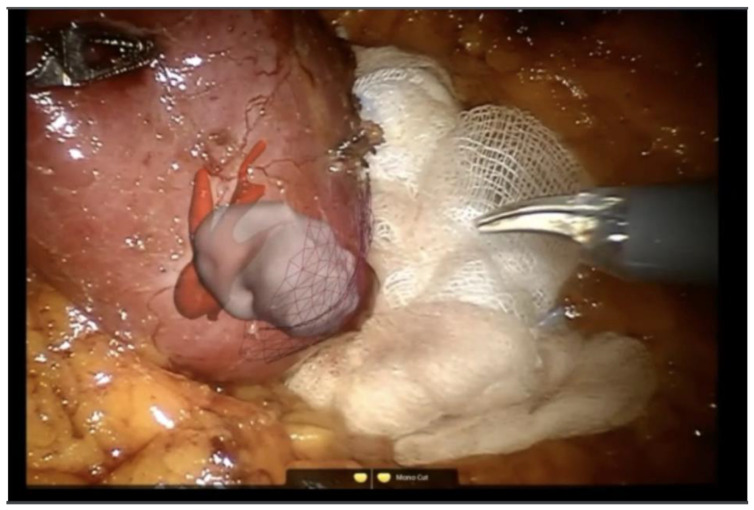
Augmented reality (AR) for partial nephrectomy (robotic surgery). Renal arteries (red) and renal tumor (grey-white) (EnCoV, SurgAR, Clermont-Ferrand, France).

**Figure 9 sensors-22-04918-f009:**
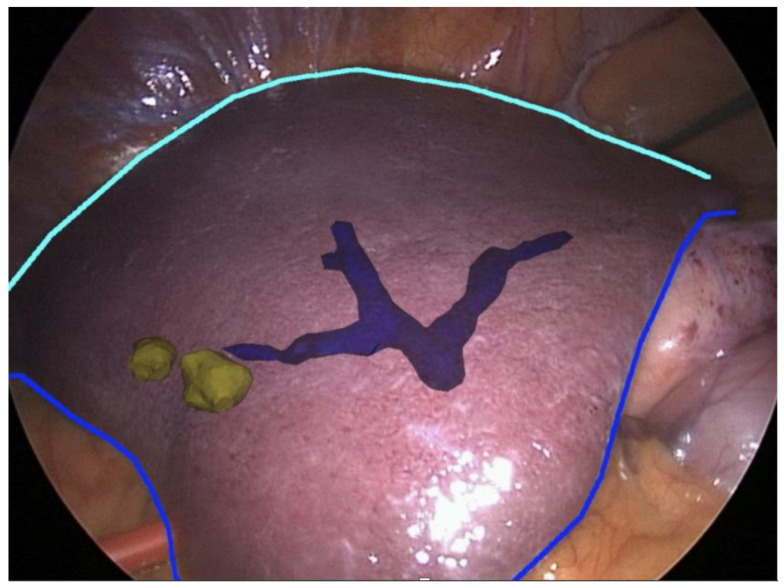
Augmented reality (AR) for partial hepatectomy. Intra-hectic portal vein (blue) and hepatic tumors (yellow) (EnCoV, SurgAR, Clermont-Ferrand, France).

## Data Availability

Not applicable.

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
