# Peer review of "The Advances in Computer Vision That Are Enabling More Autonomous Actions in Surgery: A Systematic Review of the Literature"

_sensors, 2022, doi:10.3390/s22134918_

Round 1

Reviewer 1 Report

This paper reviews the current obstacles in the creation of computer vision systems regarding autonomous actions during surgery. In addition, the authors present a discussion carried out by surgeons about the most effective way toward more advanced autonomy in surgery concerning the distance between surgeons and their patients.

 The paper organization needs a revision. On the other hand, the topic is relevant and sounds interesting. However, the main contribution is not clear. The obstacles found to computer vision systems creation, applied to autonomous actions during surgery, are not fully described.

Finally, I have some comments about your work:

1) I recommend a restructuration of the manuscript. It could help to read it. For example, Sections 1 to 10 could be added to a new related work section, and Sections 11 to 14 could be themselves a new Section.

 2) Are figures 3 to 8 from your authoring? If not, could you add the references and take care of the copyright permissions?

 3) The manuscript is hard to read. Is it possible to use some tables to organize the information?

 4) The Discussion section needs to focus on listing the current obstacles in the creation of computer vision systems regarding autonomous actions during surgery, the analysis of sound or audio approaches could be moved as a new subsection after Artificial Intelligence Surgery Section.

 5) There are some mistakes in the text and the references:

 Section 11: For example, Consider the need

Section 12: robotic surgery can be registered and analyzed_[ 51].

Reference 6: (Acomparisonofcomplete<i>vs</i>.

Reference 8: AnnalsofGastroenterologicalSurgery.n/a(n/a).

Author Response

The requested changes were made.

The end of the introduction was changed and a Methods and Results section were added to help better define the search criteria for the review.

This review article will address the current advances made in the field of computer vision (CV) as it pertains to autonomous actions during surgery, additionally, it will discuss the existential crisis for surgeons as to whether or not the safest and most effective way towards more advanced autonomy in surgery should require less or more physical distance between surgeons and their patients. Fundamentally, should we be striving for true robotic autonomy during interventional procedures, are collaborative robots (cobots) the safest way forward, or will it be a combination of both? Although robots and computers may be able to effectively “see” through traditional concepts of CV such as instrument priors and motion analysis to create the 3rd dimension, non-visual data points such as the interpretation of audio signals and light intensity readings may also be used. This review, is a follow-up to a previously published article that, in addition to CV, also expanded upon the other pillars of AI such as machine learning (ML), deep learning (DP) and natural language processing as they pertain to autonomous actions in surgery[1].

  1. METHODS

Search terms included combinations of autonomous actions in surgery/autonomous surgery/autonomous surgical robots with CV, ML, DL, augmented reality (AR), segmentation, phase recognition or navigation. Additional search terms included less traditional concepts of CV such as instrument priors and audio-haptics. Articles discussing autonomous actions in interventional healthcare disciplines were then sought. The term "surgery" was utilized in the manuscript to encompass the fields of surgery, endoscopy and interventional radiology. The fields of surgery studied included open and minimally invasive approaches. Minimally invasive approaches were defined as encompassing endoscopic and robotic-assisted approaches. Endoscopic approaches were defined as including, but not limited to laparoscopic, thorascopic-assisted and arthroscopic procedures. Only studies done or meant for human subjects were considered. Studies that were not in english were excluded. Virtual reality (VR) was not included in this review as it was not considered precise enough in its current form to offer safe autonomous actions during interventions.

Articles that were included were chosen and approved by the first 2 authors (A.G. and V.G.). This review article is meant for medical doctors that perform interventional procedures. The review attempts to familiarize doctors with the basic concepts of artificial intelligence with an emphasis on the most relevant advances in CV that are enabling more autonomous actions during interventional procedures. Review articles and studies about simulating interventions were excluded.

  1. RESULTS

This review article was registered on PROSPERO on June 6, 2022 an dis pending approval. The PRISMA checklist was used to format and organize this article. A total of 1687 articles were identified by searching PUBMED, GOOGLE SCHOLAR and REASEARCHGATE (Figure 1).

A total of 1,499 were off topic and were excluded. An additional 67 were duplicates. Of 121 articles initially identified, an additional 33 were not in the english language. Of 88 articles remaining, 3 could not be retrieved. Of these an additional 13 were found not to describe an autonomous action, leaving 72 articles for this review.

2) Are figures 3 to 8 from your authoring? If not, could you add the references and take care of the copyright permissions?

These figures are mine.

3) The manuscript is hard to read. Is it possible to use some tables to organize the information?

A PRISMA Flow diagram was added to the manuscript, please see new figure 1.

Extensive revisions were made throughout the manuscript.

4) The Discussion section needs to focus on listing the current obstacles in the creation of computer vision systems regarding autonomous actions during surgery, the analysis of sound or audio approaches could be moved as a new subsection after Artificial Intelligence Surgery Section.

Please see the extensive changes above that were done to address this, additionally,

the title of the article was changed to better reflect the contents of this review and now reads :

The Advances in Computer Vision That Are Enabling More Autonomous Actions in Surgery : How Hard is it for the Computer to See During a Procedure?

The abstract was also changed and now reads :

Abstract: This is a review focused on advances and current limitations of computer vision (CV) and how CV can help us get to more autonomous actions in surgery. It is a follow-up article to one that we previously published in Sensors entitled, “Artificial Intelligence Surgery : How Do We Get to Autonomous Actions in Surgery?” As opposed to that article that also discussed issues of Machine Learning, Deep Learning and Natural Language Processing, this review will delve deeper into the field of CV. Additionally, non-visual forms of data that can aid computerized robots in the performance of more autonomous actions, such as instrument priors and audio haptics, will also be highlighted. Furthermore, the current existential crisis for surgeons, endoscopists and interventional radiologists regarding more autonomy during procedures will be discussed. In summary, this paper will discuss how to harness the power of CV to keep doctors who do interventions in the loop.

5) There are some mistakes in the text and the references:

Section 11: For example, Consider the need

Section 12: robotic surgery can be registered and analyzed_[ 51].

Reference 6: (Acomparisonofcomplete<i>vs</i>.

Reference 8: AnnalsofGastroenterologicalSurgery.n/a(n/a).

All of these were corrected. Thank you.

Additionally, the  PRISMA guidelines were added. Please see below.

  1. METHODS

Search terms included combinations of autonomous actions in surgery/autonomous surgery/autonomous surgical robots with CV, ML, DL, augmented reality (AR), segmentation, phase recognition or navigation. Additional search terms included less traditional concepts of CV such as instrument priors and audio-haptics. Articles discussing autonomous actions in interventional healthcare disciplines were then sought. The term "surgery" was utilized in the manuscript to encompass the fields of surgery, endoscopy and interventional radiology. The fields of surgery studied included open and minimally invasive approaches. Minimally invasive approaches were defined as encompassing endoscopic and robotic-assisted approaches. Endoscopic approaches were defined as including, but not limited to laparoscopic, thorascopic-assisted and arthroscopic procedures. Only studies done or meant for human subjects were considered. Studies that were not in english were excluded. Virtual reality (VR) was not included in this review as it was not considered precise enough in its current form to offer safe autonomous actions during interventions.

Articles that were included were chosen and approved by the first 2 authors (A.G. and V.G.). This review article is meant for medical doctors that perform interventional procedures. The review attempts to familiarize doctors with the basic concepts of artificial intelligence with an emphasis on the most relevant advances in CV that are enabling more autonomous actions during interventional procedures. Review articles and studies about simulating interventions were excluded.

  1. RESULTS

This review article was registered on PROSPERO on June 6, 2022 an dis pending approval. The PRISMA checklist was used to format and organize this article. A total of 1687 articles were identified by searching PUBMED, GOOGLE SCHOLAR and REASEARCHGATE (Figure 1).

A total of 1,499 were off topic and were excluded. An additional 67 were duplicates. Of 121 articles initially identified, an additional 33 were not in the english language. Of 88 articles remaining, 3 could not be retrieved. Of these an additional 13 were found not to describe an autonomous action, leaving 72 articles for this review.

So, it is not clear how the literature was chosen. In the Augmented Reality (AR) section, this work is, for example, missing, where CV is used to detect a patient (face) for head and maxillofacial surgery:

https://link.springer.com/article/10.1007/s10278-019-00272-6

References

The references need to be checked, e.g. (name):

  1. A., D., et al., Laparoscopic task recognition using Hidden Markov Models. Stud Health Technol Inform, 2005;. 111 ( p. 15 22.

vs

  1. Rosen, J., et al., Markov modeling of minimally invasive surgery based on tool/tissue interaction and force/torque signatures for evaluating surgical skills. IEEE Trans Biomed Eng, 2001. 48 ( p. 579 91.

These were corrected thank you. Other errors in the references were also corrected. Thank you.

  1. Dosis, A., et al., Laparoscopic task recognition using Hidden Markov Models. Stud Health Technol Inform. 111: p. 115-22.

  1. Rosen, J., et al., Markov modeling of minimally invasive surgery based on tool/tissue interaction and force/torque signatures for evaluating surgical skills. IEEE Trans Biomed Eng, 2001. 48(5): p. 579-91.

Bibliography

  1. Gumbs, A.A., et al., Artificial Intelligence Surgery: How Do We Get to Autonomous Actions in Surgery? Sensors (Basel), 2021. 21(16).

Reviewer 2 Report

This paper aims for providing a review on advances and current limitations of computer vision for autonomous actions in surgery. This is a very interesting topic, and would provide insights for the surgery guidance field. However, the comments:

1) the structure of the paper is not well organized. 15 sub-chapters are used in parallel, without contextual transit. It is not easy to get the points of those paragraphs and chapters

2)there is no conclusion for the work, so 'How hard is it for the computer to see'?

3) sub-chapter 9, instance/video/surgical segmentation, why put them together? they are not the same thing.

4) sub chapter8,  the four phases of semantic segmentation, what's the reference for them?

5) 'AI has become a reality thanks to advances in ML, which is a form of supervised learning. DL is a type of ML that is unsupervised.' This is not correct.

Reviewer 3 Report

The work presents a review focused on advances and current limitations of computer vision (CV) and how CV can help to get to more autonomous actions in surgery. A systematic review, following the PRISMA guidelines, would be nice. So, it is not clear how the literature was chosen. In the Augmented Reality (AR) section, this work is, for example, missing, where CV is used to detect a patient (face) for head and maxillofacial surgery:

https://link.springer.com/article/10.1007/s10278-019-00272-6

References

The references need to be checked, e.g. (name):

51. A., D., et al., Laparoscopic task recognition using Hidden Markov Models. Stud Health Technol Inform, 2005;. 111 ( p. 15 22.

vs

52. Rosen, J., et al., Markov modeling of minimally invasive surgery based on tool/tissue interaction and force/torque signatures for evaluating surgical skills. IEEE Trans Biomed Eng, 2001. 48 ( p. 579 91.

Reviewer 4 Report

This paper provides a literature review for computer vision and other AI techniques applied to surgery. It covers many surgery fields and how images and non-vision data can introduce more autonomous actions in surgery. Multiple major CV tasks have been discussed, like object detection, semantic segmentation, AR, VR, and reinforcement learning. It's very excited to see so many medical applications of CV and AI listed in this paper. Generally this paper is well written and interesting to our readers. 

Some concerns are:

1. In "15. Discussion - DL is a type of ML that is unsupervised" which is not accurate. DL is a kind of ML approach using multi-layer neural networks, but this kind of models can be used in both supervised and unsupervised tasks. 

2. In the end of section 9. Instance/Video/Surgical Segmentation, more references [1] can be included for "analyzing both temporal and spatial signals". 

3. In addition to CV, this paper also introduced some techniques using other types of sensors, like audio and video data. More reference can be included for sensor fusion, i.e. using multimodal data for decision making. 

4. Some typos, such as before reference [51] and [52], there should be a space. 

5. In section 6, "YOLO is considered among the fastest object recognition methods", is YOLO still the state-of-the-art algorithm for object recognition?

6. Some figures are too wide to fit the context. For example figure 1,2,3 can be clipped, resized, or aligned to the center. 

[1]. Correlative Mean-Field Filter for Sequential and Spatial Data Processing, in the Proceedings of IEEE International Conference on Computer as a Tool (EUROCON), Ohrid, Macedonia, July 2017

Round 2

Reviewer 1 Report

The manuscript was improved from its last version. The authors addressed all requirements. Thank you.

 I recommend performing a new revision to correct a few details and correct some typos.

Author Response

Additional typos, repetitive words and grammatical errors were found and corrected. Thank you.

Additionally a new reference was added.

Reviewer 2 Report

I appreciate the authors addressing my comments. The figures in the paper can be more clear, and some typos should be corrected. 

Author Response

Some typos, repetitive words and other grammatical errors were found and fixed.

Additionally, another reference was added.

Thank you

Reviewer 3 Report

I thank the authors for addressing my points. From my point of view, the paper is now much stronger. The authors could consider adding “systematic review” to the title, which would attract more readers/citations, e.g.:

The Advances in Computer Vision that are enabling more autonomous Actions in Surgery: A systematic Review of the Literature

However, I will endorse the paper for publication.

Author Response

Thank you for the endorsement, the title was changed as suggested.

Our regards